# Vascular and Liver Homeostasis in Juvenile Mice Require Endothelial Cyclic AMP-Dependent Protein Kinase A

**DOI:** 10.3390/ijms231911419

**Published:** 2022-09-27

**Authors:** Pavel I. Nedvetsky, Ivo Cornelissen, Thomas Mathivet, Claire Bouleti, Phalla Ou, Pieter Baatsen, Xiaocheng Zhao, Frans Schuit, Fabio Stanchi, Keith E. Mostov, Holger Gerhardt

**Affiliations:** 1VIB-KU Leuven Center for Cancer Biology, VIB, 3000 Leuven, Belgium; 2Department of Oncology, KU Leuven, 3000 Leuven, Belgium; 3Medical Cell Biology, Medical Clinic D, University Hospital Münster, 48149 Münster, Germany; 4Bordeaux Institute of Oncology, BRIC U1312, INSERM, Université de Bordeaux, 33000 Bordeaux, France; 5Cardiology Department, Assistance Publique-Hôpitaux de Paris, Bichat Hospital, 75018 Paris, France; 6Department of Radiology, Assistance Publique-Hôpitaux de Paris, Bichat Hospital, 75018 Paris, France; 7VIB-KU Leuven Center for Brain and Disease Research, EM-Platform at the VIB Bio Imaging Core, 3000 Leuven, Belgium; 8Gene Expression Unit, KU Leuven, 3000 Leuven, Belgium; 9Department of Anatomy, University of California San Francisco, San Francisco, CA 94158, USA; 10Max-Delbrück Center for Molecular Medicine in the Helmholtz Association (MDC), 13125 Berlin, Germany; 11DZHK (German Center for Cardiovascular Research), Partner Site Berlin, 10785 Berlin, Germany; 12Berlin Institute of Health (BIH), 10178 Berlin, Germany

**Keywords:** angiogenesis, edema, liver sinusoidal endothelium

## Abstract

During vascular development, endothelial cAMP-dependent protein kinase A (PKA) regulates angiogenesis by controlling the number of tip cells, and PKA inhibition leads to excessive angiogenesis. Whether this role of endothelial PKA is restricted to embryonic and neonatal development or is also required for vascular homeostasis later on is unknown. Here, we show that perinatal (postnatal days P1–P3) of later (P28–P32) inhibition of endothelial PKA using dominant-negative PKA expressed under the control of endothelial-specific Cdh5-CreERT2 recombinase (dnPKA^iEC^ mice) leads to severe subcutaneous edema, hypoalbuminemia, hypoglycemia and premature death. These changes were accompanied by the local hypersprouting of blood vessels in fat pads and the secondary enlargement of subcutaneous lymphatic vessels. Most noticeably, endothelial PKA inhibition caused a dramatic disorganization of the liver vasculature. Hepatic changes correlated with decreased gluconeogenesis, while liver albumin production seems to be unaffected and hypoalbuminemia is rather a result of increased leakage into the interstitium. Interestingly, the expression of dnPKA only in lymphatics using Prox1-CreERT2 produced no phenotype. Likewise, the mosaic expression in only endothelial subpopulations using Vegfr3-CreERT2 was insufficient to induce edema or hypoglycemia. Increased expression of the tip cell marker ESM1 indicated that the inhibition of PKA induced an angiogenic response in the liver, although tissue derived pro- and anti-angiogenic factors were unchanged. These data indicate that endothelial PKA is a gatekeeper of endothelial cell activation not only in development but also in adult homeostasis, preventing the aberrant reactivation of the angiogenic program.

## 1. Introduction

The vasculature comprises a highly-branched tubular network of blood and lymphatic vessels, transporting blood cells and plasma containing nutrients, hormones, gases, and other molecules through the body. Moreover, endothelial cells that line the inner surface of all blood vessels contribute to the regulation of tissues and organs through the production of signaling (angiocrine) factors [1,2,3,4]. Most of the vessels in the body are formed during development by a process called angiogenesis, describing the formation of new vessels from preexisting ones. While endothelial cells are highly motile and proliferative during angiogenesis, in healthy and functional vessels, they are mostly quiescent. However, they preserve the ability to return to a highly motile and proliferative state if challenged by pro-angiogenic factors (e.g., during wound healing, inflammation or tumor vascularization).

In addition to maintaining quiescence, endothelial cells acquire organ-specific characteristics. Endothelial cells in various organs differ from each other in their morphology, subcellular organization, susceptibility to hormones and growth factors as well as in their protein expression profiles [3,5]. A perfect example for such organ-specific specialization of blood vessels and endothelial cells is the vasculature of the liver. While the large hepatic vessels are similar to large blood vessels in other organs, the smallest blood vessels of the liver, the sinusoids, are very different from capillaries that make up the smallest blood vessels in most other organs. Liver sinusoidal endothelial cells (LSEC) are fenestrated, lack well-organized basement membrane and express several unique marker proteins [6].

The erroneous or overshooting activation of angiogenesis is a hallmark of several pathologies, including cancer and rheumatoid arthritis. Conversely, a perturbed angiogenic response may lead to insufficient vascularization during wound healing and tissue regeneration. While it is clear that the balance of pro- and anti-angiogenic factors is essential to maintain endothelial cell quiescence, much less is known about intrinsic factors fine-tuning the response of endothelial cell to these factors.

Cyclic AMP (cAMP)-dependent protein kinase A (PKA) is a ubiquitously expressed protein kinase, which is activated by the second messenger cAMP. By phosphorylating its numerous substrates, PKA is involved in regulating the function and homeostasis of virtually every tissue and organ [7,8,9]. In endothelial cells, PKA has been shown to regulate endothelial barrier function [1,10,11,12,13], response to hemodynamic forces [2,3,4] and the production of nitric oxide [8,9] as well as inhibit angiogenesis [1,10,11,12,13]. We previously demonstrated that endothelial PKA is required to prevent excessive hypersprouting during mouse development [1]. While the constitutive inhibition of endothelial PKA is embryonically lethal, mice survived until adulthood when endothelial PKA was inhibited postnatally. This leaves open the question of whether endothelial PKA is only required during development of the vascular system or also for vascular homeostasis. Our present data demonstrate that PKA activity is essential for vascular homeostasis and to prevent pathogenic vascular remodeling. Most strikingly, the inhibition of endothelial PKA led to disorganization of the hepatic vasculature with detrimental metabolic consequences—mice developed edema and hypoglycemia, causing premature death.

## 2. Results

### 2.1. Inhibition of Endothelial PKA in Mice Results in Edema and Premature Death

In our previous study, we showed that the genetic inhibition of endothelial PKA during development leads to excessive angiogenesis [1]. While the constitutive inhibition of endothelial PKA was embryonically lethal, mice survived for at least one month if endothelial PKA was inhibited postnatally. Moreover, the initially severely perturbed retinal vasculature normalized to a significant extent in these mice [1], raising the question of whether endothelial PKA is required only during vascular development and is dispensable for vascular homeostasis thereafter. Here, we assessed the effects of inhibiting endothelial PKA in neonatal and juvenile mice by analyzing dnPKA^iEC^ mice (floxed dominant negative PKA mice [14] crossed with the endothelial-specific Cdh5-CreERT2 line [15]). Previous studies have demonstrated that the Cre-dependent expression of dnPKA resulted in a 50% reduction in cAMP-induced PKA activity in the affected tissues [1,14].

When Cre-dependent recombination was induced during the first 3 days after birth, all dnPKA^iEC^ mice survived until the age of one month but later died prematurely with more than 50% of mice dying between one and three months of age (Figure 1A; here and below, we describe mice expressing corresponding Cre recombinase transgene but lacking the dnPKA allele as control mice). Mutant mice had a broader weight distribution—with some mice failing to gain weight properly and about half of them gaining more weight than the control littermates (Figure 1B). Interestingly, this weight gain in dnPKA^iEC^ mice was not due to obesity and fat deposition but due to severe subcutaneous edema (Figure 1C–E and see below). Mice developed a thick gelatinous layer underneath the skin, which accounted for almost all of the weight difference between the mutant and the control mice (Appendix A) and was clearly visible on MRI images (Figure 1E and Appendix A). Interestingly, the accumulation of Evans blue was not increased but rather decreased in all other organs tested (Figure 1F), indicating that the endothelial barrier function in inner organs was mostly preserved. 

The inhibition of endothelial PKA during early postnatal development was not required for edema development, since the injection of dnPKA^iEC^ mice with tamoxifen after weaning (at 4 weeks of age) resulted in rapid edema development as well. Within a month, dnPKA^iEC^ mice gained about 75% of their weight and developed subcutaneous edema, while the control littermates gained only about 25% during the same period and had no sign of edema (Figure 1G). These data indicate that endothelial PKA is required not only during developmental angiogenesis but also plays an important role in vascular homeostasis in juvenile mice. For all further experiments, except when indicated, dnPKA^iEC^ mice were injected with tamoxifen at the age of 4 weeks to minimize potential influences of developmental effects.

### 2.2. Differential Response to PKA Inhibition in Different Vascular Beds

Since severe subcutaneous edema and vascular leakage indicate defects of cutaneous blood vessels, we analyzed the morphology of cutaneous blood vessels in the dorsal skin of dnPKA^iEC^ mice. Surprisingly, we found no difference in vessel shape or density between dnPKA^iEC^ mice and their control littermates (Figure 2A). These results were unexpected, given the massive accumulation of vascular leakage tracer in the subcutaneous space of dnPKA^iEC^ mice (Figure 1D). However, analysis of the vasculature in the inguinal fat pad revealed major changes in vascular morphology. The inguinal fat pads of both male and female dnPKA^iEC^ mice were penetrated by a highly branched vascular network, which was not visible in the fat pads of control mice (Figure 2B). The same was true for the thoracic fat pads (Figure 2C). Staining of inguinal fat pads with isolectin B4 confirmed a much higher density of blood capillaries in dnPKA^iEC^ mice compared to the control littermates (Appendix A) and indicated that the inhibition of endothelial PKA leads to the angiogenic activation of endothelial cells, which might be restricted to specific vascular beds without affecting others.

### 2.3. Inhibition of Endothelial PKA in Lymphatics Is Not Sufficient to Induce Edema

In addition to blood vessel hypersprouting in fat pads, the inhibition of endothelial PKA led to a dramatic enlargement of cutaneous lymphatic vessels in the ventral abdominal skin (Figure 3A,A’,B), indicating that both blood vessels and lymphatics are affected in dnPKA^iEC^ mice and might contribute to the development of the edema.

Cadherin-5 (Cdh5), which was used to drive Cre expression in dnPKA^iEC^ mice, is expressed in all endothelial cells [16], and thus, dnPKA is expressed both in blood and lymphatic endothelial cells. To test whether the inhibition of PKA in lymphatic endothelial cells was sufficient to induce the enlargement of lymphatics and edema, we induced the expression of dnPKA with lymphatic-specific Cre-lines, Prox1-CreERT2 [17] and Vegfr3-CreERT2 [18].

When injected with tamoxifen during postnatal days 1 to 3, neither dnPKA:Prox1-CreERT2 nor dnPKA:Vegfr3-CreERT2 mice developed any signs of edema at least until the age of four months (Figure 3C,D). To confirm successful recombination, mice carrying the mT/mG Cre-reporter were used [19]. We confirmed that lymphatic vessels both in the diaphragm (Figure 4A) and ventral abdominal skin (Figure 4B) of the Vegfr3-CreERT2 and dnPKA:Vegfr3-CreERT2 mice expressed GFP, demonstrating successful recombination. Importantly, no enlargement of lymphatic vessels was observed in these mice (Figure 4A,B). This indicates that lymphatic effects observed in dnPKA^iEC^ mice are secondary to blood vessel defects, and they are a consequence of the extensive subcutaneous edema rather than a cause of it.

### 2.4. Changes in Blood Parameters Indicate Hepatic Defects in dnPKA^iEC^ Mice

Severe edema may contribute to the premature death of dnPKA^iEC^ mice (Figure 1A), e.g., due to cardiovascular and hemodynamic complications. However, when analyzing dnPKA^iEC^ mice injected with tamoxifen during the first three days after birth, we observed that mice that failed to gain weight died earlier than the mice that developed edema. This suggests that it is not the edema itself but rather some other perturbation(s) that contribute to the premature lethality in dnPKA^iEC^ mice.

To identify such potential perturbation(s), we examined several standard plasma chemistry parameters in dnPKA^iEC^ and control mice. Levels of LDH, creatinine, creatine kinase, ALT, AST and ALP were not significantly changed (Figure 5A–F), which is consistent with the absence of gross damage to the kidney, skeletal muscle or liver. However, we observed decreased levels of serum albumin (Figure 5G) that correlated with the severity of edema (Figure 6A). Moreover, plasma glucose was decreased in dnPKA^iEC^ mice (Figure 5H). The hyperactivity of pancreatic beta cells seemed unlikely to be responsible for hypoglycemia, as plasma insulin levels were not increased in dnPKA^iEC^ mice (Figure 6B). We hypothesized that both hypoalbuminemia and hypoglycemia may be a consequence of impaired liver function.

In accordance with this hypothesis, much slower and more modest edema (Figure 6C) and no hypoglycemia (Figure 6D) were observed in mice expressing dnPKA under control of Pdgfb-iCre, which is not expressed in hepatic blood vessels [20]. For comparison, while no differences between the control and Pdgfb-iCre:dnPKA mice have been observed until about 50 days of age (Figure 6D), about 50% of dnPKA^iEC^ mice died by this timepoint (Figure 1A). We have previously demonstrated that during early postnatal development, the induction of dnPKA expression by Pdgfb-iCre recapitulated the retinal hypersprouting phenotype observed in dnPKA^iEC^ mice, Ref. [1] suggesting that the induction of dnPKA by the Pdgfb-iCre was able to efficiently inhibit PKA activity in endothelial cells where it is expressed. Based on these data, we hypothesized that hepatic vascular defects play an important role in the edema and hypoglycemia observed in dnPKA^iEC^ mice. Interestingly, in dnPKA:Vegfr3-CreERT2 mice that should express dnPKA not only in lymphatic but also in LSECs [18,21], no hypoglycemia was observed (Figure 3E). Since Cdh5-CreERT2 induces recombination in a much broader spectrum of liver endothelial cells than the Vegfr3-CreERT2 (Appendix A), we hypothesize that the severe effects observed in dnPKA^iEC^ mice require a broad inhibition of PKA in most or all endothelial cells of the liver, while PKA inhibition in a more restricted cell population was insufficient. Moreover, extra-hepatic vascular defects under inhibition of endothelial PKA may also be an important factor contributing to the development of edema and hypoglycemia.

### 2.5. Gluconeogenesis but Not Albumin Expression Is Perturbed in dnPKA^iEC^ Mice

While edema is a frequent consequence of vascular defects, hypoglycemia is more difficult to explain by the intrinsic defects of endothelial cells. Since our data suggest involvement of the liver in the development of hypoglycemia, we tested whether metabolic functions of the liver were impaired in dnPKA^iEC^ mice. Gluconeogenesis, the generation of glucose from non-carbohydrate carbon sources, is an important metabolic process occurring in the liver, which maintains constant levels of blood glucose under conditions of limited energy supply or increased energy consumption. We wondered whether gluconeogenesis may be perturbed in dnPKA^iEC^ mice and performed a pyruvate tolerance test to address this. Mice were fasted overnight and injected with a bolus of pyruvate to fuel gluconeogenesis that quickly converts pyruvate into glucose. As expected, in control mice, the injection of pyruvate resulted in a rapid and transient increase in the blood glucose level, indicating the conversion of pyruvate into carbohydrates. Importantly, this transient increase in blood glucose level was blunted in dnPKA^iEC^ mice (Figure 7A), suggesting that gluconeogenesis was indeed perturbed in these mice. Importantly, if mice were instead injected with a bolus of glucose, there was no difference in the dynamics of blood glucose changes between control and dnPKA^iEC^ mice (Figure 3B), indicating that neither the distribution of intraperitoneal injected tracer nor the ability of mice to respond to an increase in blood glucose was affected in dnPKA^iEC^ mice. 

To study whether reduced production of albumin by the liver is responsible for hypoalbuminemia, we determined the hepatic expression of albumin by QPCR and Western blotting. While the level of albumin mRNA was increased (Figure 8A), the albumin protein level in the liver of dnPKA^iEC^ mice was unchanged compared to the control mice (Figure 8B,D). Thus, hypoalbuminemia is not a consequence of reduced albumin production by the liver. Rather, drainage of the albumin into the subcutaneous space, as indicated by albumin-binding tracer Evans blue, seems to be the cause for hypoalbuminemia. Taken together, these data demonstrate that hepatic malfunctions are responsible for hypoglycemia but not hypoalbuminemia in dnPKA^iEC^ mice.

We also analyzed the expression levels of three key gluconeogenic enzymes, glucose-6-phosphatase (*G6pc*), pyruvate carboxylase (*Pcx1*) and phosphoenolpyruvate carboxykinase (Pck1). Neither the mRNA levels of *G6pc* or *Pcx1* (Figure 8A) nor the protein levels of Pck1 (Figure 8C,D) were significantly changed in dnPKA^iEC^ mice compared to control mice.

### 2.6. Inhibition of Endothelial PKA Leads to Disorganization of Hepatic Vasculature

The abnormalities in several plasma parameters and the pyruvate tolerance test suggest that hepatic function is affected in dnPKA^iEC^ mice. Indeed, the expression of the catalytic subunit of the essential enzyme involved in gluconeogenesis G6pc was consistently reduced (Figure 8). In addition, Pcx1, a redox regulator in the liver and kidney, showed reduced expression. Critical homeobox nuclear factor family members that are normally downregulated in severe liver disease such as Hnf4a and Hnf1b were, however, unchanged. Therefore, we analyzed whether the inhibition of endothelial PKA resulted in histological abnormalities of the liver. To this end, we first analyzed liver slices from wild-type and dnPKA^iEC^ mice expressing the mT/mG reporter to gain information about the overall liver anatomy in dnPKA^iEC^ mice as well as the degree of Cre-mediated recombination in the hepatic vasculature. In mT/mG mice, cells without recombination express red fluorescent protein dTomato, while recombination results in the expression of green fluorescent protein (GFP) [19]. As expected, lobular organization of the liver was easily revealed by the expression of dTomato in the dnPKA negative mice with uniform sinusoids expressing GFP in virtually every endothelial cell (Figure 9). In dnPKA^iEC^ mice, a striking disorganization of the hepatic tissue with enlarged vessels was observed (Figure 9 and higher magnification in Appendix A). 

We hypothesized that the severe disorganization of the hepatic vasculature observed in dnPKA^iEC^ mice resulted in a perturbation of the hepatocyte differentiation state and function. For instance, liver zonation, the spatially organized localization of hepatocytes performing different metabolic pathways in the hepatic acinus, is essential for hepatic functions. It has previously been demonstrated that liver endothelial cells produce R-spondin-3, a Wnt ligand and important morphogen controlling liver zonation [22]. Therefore, we tested whether liver zonation was affected in dnPKA^iEC^ mice. We stained liver slices for glutamine synthetase (GS), which is a well-known zonation marker and Wnt target, the expression of which is affected by the perturbation of Wnt signaling [22,23,24]. As expected, in control mice, GS is expressed in hepatocytes residing in direct vicinity of the central vein (Figure 10, left lower panel). In dnPKA^iEC^ mice, this pericentral expression of GS was preserved (Figure 10, right lower panel). Interestingly, the staining of blood vessels with isolectin B4 and anti-endomucin antibody were not uniform throughout a liver lobule. While staining for endomucin was strongest at the central vein and the pericentral vasculature, isolectin B4 stained these areas much weaker than the periportal ones (Figure 10, upper and middle panels, respectively). In dnPKA^iEC^ mice, hepatic vessels were severely disorganized, making this spatial separation less evident (Figure 10, right panels). However, comparing endomucin and isolectin B4 staining clearly revealed that most of the enlarged vessels were positive for isolectin B4 and negative for endomucin (Figure 10, middle panels). This might be an indirect indication that vessel enlargement in dnPKA^iEC^ mice preferentially affects the periportal part of the hepatic vasculature. Alternatively, vessel enlargement may coincide with the upregulation of isolectin B4 ligands. On the other side, the downregulation of endomucin as the case for this effect seems to be unlikely, since its expression was not affected in the dnPKA^iEC^ mice (see Figure 11).

These data suggest that in dnPKA^iEC^ mice, liver (hepatocyte) zonation is preserved. Morphological changes in liver structure are manifestations of vascular disorganization rather than defects in the liver parenchyma. Additional support for this hypothesis was provided by unchanged levels of hepatocyte markers, hepatic transcription factors *Hnf1b* and *Hnf4a* and of the hepatic enzymes, glucose-6-phosphatase-α (*G6pc*) and pyruvate carboxylase (*Pcx1*), in dnPKA^iEC^ mice (Figure 8). Together with the preservation of albumin production, these data suggest that dnPKA^iEC^ mice do not suffer generalized liver damage but experienced more specific dysfunctions due to disorganization of the hepatic vasculature.

### 2.7. LSECs Preserve Their Differentiation State in dnPKA^iEC^ Mice

It is well established that the dedifferentiation of LSECs, so-called capillarization, is associated with liver pathologies (e.g., liver fibrosis) [6]. We hypothesized that the inhibition of endothelial PKA may lead to the capillarization of hepatic sinusoids and result in defects of hepatic functions. However, the expression of LSEC markers, *Gata4*, stabilin2 (*Stab2*) and *Vegfr3*, was not affected in the liver of dnPKA^iEC^ mice (Figure 11), suggesting that LSECs preserved their differentiation state under PKA inhibition. Moreover, the expression of *Lyve1*, which is downregulated during capillarization [25], was increased in dnPKA^iEC^ mice, as evident both by QPCR analysis (Figure 11) and immunostaining (Figure 12A, middle and lower panels). Finally, we analyzed whether fenestration, a hallmark of hepatic sinusoids, was lost in dnPKA^iEC^ mice. No differences in the degree of fenestration could be found between dnPKA^iEC^ mice and their control littermates (Figure 12B). Quantification of the porosity of liver sinusoids, 3.8 ± 0.3% (n = 5) for control mice and 4.1 ± 1.7% (n = 4) for dnPKA^iEC^ mice confirmed this conclusion. These results indicate that LSECs preserve their differentiation state and do not undergo capillarization but profoundly disorganize under the inhibition of endothelial PKA. 

### 2.8. Inhibition of Endothelial PKA Results in Cell-Autonomous Reactivation of the Angiogenic Program

During development, the inhibition of endothelial PKA led to excessive angiogenesis due to cell-autonomous sensitization of endothelial cells [1]. This manifested in an increase in the number of tip cells, which are endothelial cells that acquired exploratory behavior and migrate along the gradient of pro-angiogenic factors. To study whether the same cell-autonomous sensitization of endothelial cells occurs in juvenile dnPKA^iEC^ mice, we analyzed the expression of a well-known tip cell marker, Esm1 [7,8,9], in the liver of dnPKA^iEC^, since the hepatic vasculature was most prominently affected in these mice. We were unable to detect Esm1 in the liver by immunostaining, which was most likely due to low expression levels of Esm1. Therefore, we attempted to analyze the expression of *Esm1* by QPCR. Since Esm1 is specifically expressed by tip cells [7], *Esm1* transcript levels can be used as an indirect measure of tip cells. The level of *Esm1* in dnPKA^iEC^ liver was more than 20-fold higher than in control littermates (Figure 13), indicating that the inhibition of endothelial PKA in the juvenile liver results in the activation of an angiogenic program and increase/reappearance of tip cells. 

To study whether the reactivation of angiogenesis was due to the intrinsic sensitization of endothelial cells and not caused by increased pro-angiogenic stimuli, we analyzed the expression of multiple pro-angiogenic factors in dnPKA^iEC^ mice. 

QPCR analysis of liver samples revealed no differences in the levels of *Vegfa*, *Vegfc*, *Fgf2*, *Hgf* or *Dll4* between control and dnPKA^iEC^ mice (Figure 13). Since angiogenesis in the adult organism is often a consequence of inflammation [26,27], we tested the expression of several endothelial markers that are frequently increased in response to inflammatory stimuli, *Vcam1*, *Icam1* and *Sele* (E-selectin) [28]. None of these markers was significantly increased in dnPKA^iEC^ mice (Figure 13), arguing against the inflammatory activation of hepatic endothelial cells. Additionally, we measured TNFα in the blood samples of control and dnPKA^iEC^ mice as a marker of systemic inflammation. However, TNFα was at undetectable levels both in control and dnPKA^iEC^ mice. Altogether, these data indicate that the increased production of pro-angiogenic factors is unlikely to be the cause of vascular disorganization in dnPKA^iEC^ mice. Instead, we propose that vascular disorganization is rather due to intrinsic changes in endothelial cells driven by the absence of proper PKA regulation.

## 3. Discussion

We report here that endothelial PKA is essential for vascular homeostasis in juvenile mice. The selective inhibition of endothelial PKA led to excessive hypersprouting of blood vessels in fat pads, enlargement of cutaneous lymphatic vessels, and disorganization of the hepatic vasculature, resulting in edema and devastating metabolic consequences. 

Our current understanding of the role of PKA in regulating vascular development and homeostasis is rudimentary. While it has been known for several decades that PKA regulates the endothelial barrier and antagonizes inflammation-induced vascular permeability [29,30], the role of PKA in angiogenesis is just starting to emerge. Some of the first indications that PKA restrains the activity of pro-angiogenic stimuli came from the group of Judith Varner [12,31]. In these studies, angiogenesis in the chick chorioallantoic assay or in zebrafish was attenuated by the pharmacological activation of PKA [12,31]. In our previous study, we observed that the inhibition of endothelial PKA during development results in excessive angiogenesis due to over-abundant tip cell formation [1]. Thus, PKA is an important player controlling developmental angiogenesis. Now, we report that beyond its role during development, active endothelial PKA is also required to maintain vascular homeostasis. 

We hypothesize that the severe edema, hypoglycemia and premature death under the inhibition of endothelial PKA in juvenile mice relies on the same mechanism as in perinatal mice [1]—sensitization of endothelial cells toward angiogenic stimuli. Indeed, both in our present and in the previous study [1], we observed increased numbers of tip cells or expression of the tip cell marker ESM1, disorganization of blood vessels and local increase in vascular density without an increase in the levels of pro-angiogenic factors. 

Interestingly, different vascular beds seem to be unequally sensitive to PKA inhibition. Hypersprouting in fat pads and progressive disorganization of the hepatic vasculature were observed in dnPKA^iEC^ mice; however, no evidence of disorganization, hypersprouting or remodeling was evident in cutaneous vessels. While lymphatic vessels enlarged and hepatic vessels underwent progressive disorganization (data presented in this study), the retinal vasculature of dnPKA^iEC^ mice, the vascular bed where dramatic hypersprouting occurred during the early postnatal period, recovered at later stages [1]. The presence of areas with no apparent morphological changes or increase in vascular permeability (dorsal skin) and areas severely affected under PKA inhibition (liver, fat pads) is an additional indication that inhibition of PKA by itself does not activate the angiogenic program but might rather prime the vasculature to respond to pro-angiogenic stimuli. Recovery of the retinal vasculature in dnPKA^iEC^ mice, which was reported earlier [1], suggests that even endothelial cells with inhibited PKA maintain the ability to return to a “normal” state when pro-angiogenic stimulation ceased. Therefore, we propose that the most severe manifestations of PKA inhibition would be observed in tissues experiencing prolonged exposure to pro-angiogenic stimuli. For example, the disorganization of hepatic vasculature in dnPKA^iEC^ mice may be the consequence of tonic production of VEGFA by hepatocytes [32] to maintain the fenestration of the liver sinusoids [33]. In contrast, most other cells do not produce VEGFA under normal conditions in adulthood [34].

The disorganization of the hepatic vasculature in dnPKA^iEC^ mice seems to be the most severe consequence of the inhibition of endothelial PKA, as it resulted in impaired gluconeogenesis in the liver and led to severe hypoglycemia. This in turn was most likely the primary reason for the death of dnPKA^iEC^ mice. Surprisingly, the disorganization of the liver vascularization was not accompanied by dedifferentiation of LSECs or an inflammatory response, which are both common features of liver fibrosis. Moreover, neither liver zonation nor the expression of markers of hepatocyte differentiation were overtly affected in these mice. However, gluconeogenesis was impaired in dnPKA^iEC^ mice, indicating that vascular defects are responsible for the metabolic dysfunction of hepatocytes.

How disorganization of the hepatic vasculature translates into metabolic defects remains an open question. The fenestration of liver sinusoids was preserved in dnPKA^iEC^ mice, indicating that transport of the metabolites from blood to hepatocytes and vice versa should not be affected. The ratio of spleen to liver weight, which is usually increased if liver perfusion is impaired, was unchanged in dnPKA^iEC^ mice. Although rather speculative at the present point, we favor endothelium-derived humoral factors being responsible for metabolic defects in dnPKA^iEC^ mice.

Endothelial cells produce multiple angiocrine factors that regulate differentiation, zonation and the functional state of other liver cells [6]. It is therefore plausible that endothelial cells affected by PKA inhibition produce a different set of factors and thus may negatively affect gluconeogenesis in hepatocytes. For instance, we show here that the *Esm1* level is increased in the livers of dnPKA^iEC^ mice. ESM1 is a proteoglycan that has been shown to facilitate HGF-dependent signaling [35]. In turn, HGF negatively regulates gluconeogenic enzymes [36,37]. In addition, another angiocrine factor, nitric oxide, is known to inhibit gluconeogenesis [38].

Several pathological conditions (e.g., cancer, arthritis) are accompanied by angiogenesis. In these conditions, inflammatory stimuli and immune cells play an important role in the activation of endothelial cells, triggering angiogenesis. The inhibition of endothelial PKA does not seem to provoke inflammation or the over-production of pro-angiogenic factors. Instead, PKA acts as an endothelial cell-intrinsic brake, preventing their overactivation when exposed to a pro-angiogenic environment. Whether such a reactivation of the angiogenic program contributes to liver diseases in humans remains unclear. Rather, the changes in the hepatic vasculature under the inhibition of endothelial PKA indicate the important physiological role of PKA activity in maintaining the homeostasis and patterning of liver blood vessels.

Several important aspects of PKA effects in endothelial cells remain unknown and require further studies. First and foremost, the molecular mechanisms of PKA effects on the angiogenic state of endothelial cells are not known. Since PKA is a pleiotropic kinase with a vast number of potential targets, it is more than likely that multiple mechanisms are integrated into anti-angiogenic effects of PKA. Several of these possible mechanisms were identified by previous studies. PKA has been shown to inhibit the migration of endothelial cells, angiogenesis and vascular permeability by phosphorylation and activation of a negative Src regulator, Csk [12]. Since Src plays an important role in the regulation of angiogenesis and vascular permeability by VEGF signaling [39], the inhibition of PKA and failure to restrict Src activity may result in increased vascular permeability and the reactivation of angiogenesis or vascular remodeling. Importantly, PKA can curb signaling by VEGF and other angiogenic growth factors via several other mechanisms. For instance, PKA has been shown to inhibit VEGF- and FGF2-dependent signaling on the level of Raf-1 [40]. Moreover, growth factor signaling is not the only pathway that may be responsible for anti-angiogenic effects of PKA. Previously, we demonstrated that endothelial PKA inhibits autophagy by phosphorylation of the autophagy protein ATG16L1 and inhibition of autophagy partially rescued hypersprouting during neonatal angiogenesis in dnPKA^iEC^ mice [41]. However, whether the regulation of autophagy is also important for the PKA-dependent maintenance of vascular homeostasis and which additional mechanisms are involved in this process remain unclear.

Another unanswered question raised by the findings of our study is the contribution of specific vascular beds to the overall effects observed on the organ or organismal level. The hepatic vasculature was drastically perturbed in dnPKA^iEC^ mice but unaffected in the dnPKA:Vegfr3-CreERT2 mice, despite the reported expression of VEGFR3 in LSECs [18] and Vegfr3-CreERT2-induced recombination in these cells. Notably, however, in our hands, Vegfr3-CreERT2 induced variable and highly mosaic recombination in the liver, suggesting that residual PKA activity in some endothelial cells may suffice to control homeostasis. We thus hypothesize that the severe abnormalities observed in the dnPKA^iEC^ mice require the inhibition of endothelial PKA en masse to overwhelm the homeostasis regulation and provoke vascular and metabolic defects.

In summary, our present study demonstrates that endothelial PKA is an important factor of vascular homeostasis, and its inhibition results in severe tissue-specific disorganization of the vasculature and, consequently, defects in the metabolic function of the liver.

## 4. Materials and Methods

### 4.1. Animal Experimentation

All experimental animal procedures were approved by the Institutional Animal Care and Research Advisory Committee of the University of Leuven (KU Leuven; Ethical committee declaration P249/2014) and performed according to the European guidelines. The following mouse strains were used: Prkar1atm2Gsm [14], Tg(Cdh5-cre/ERT2)1Rha [15], Gt(ROSA)26Sortm4(ACTB-tdTomato,-EGFP)Luo/J [19], Tg(Pdgfb-icre/ERT2)1Frut [20], Vegfr3-CreERT2 [18] and Prox1-CreERT2 [17]. All animals used in the experiments were mixed N/FVB × C57/Bl6 mice with at least 75% N/FVB. All comparisons were made between littermates only. Except for the degree of edema (which was more severe in dnPKA^iEC^ males than females), all the effects described affected both females and males to a similar extent.

Cre-dependent recombination was induced by intraperitoneal injection of tamoxifen (diluted in corn oil) either at postnatal day 1 to 3 (P1–3) or around 4 weeks of age (P28–32). Pups (P1–3) were injected with 50 µg tamoxifen per day; juvenile mice (P28–32) were injected with 40 µg/g tamoxifen per day. Control mice were defined as one lacking floxed dnPKA allele (having both wt Prkar1a alleles). Control mice carried other transgenes (e.g., CreERT2 and mTmG when indicated) similarly to mutant littermates and were treated with tamoxifen in the same way as their mutant littermates. Mice carrying both Tg(Cdh5-cre/ERT2)1Rha transgene and floxed Prkar1atm2Gsm allele were defined as dnPKA^iEC^ mice. 

For analysis of vascular leakage, mice were anesthetized and injected intravenously with 100 µL 1% Evans blue solution (Sigma, Taufkirchen, Germany). One hour after injection, mice were sacrificed, and indicated organs were isolated. Organs were weighed and incubated in formamide solution at 56 °C for 24 h to extract the dye. The absorbance of the solution was measured with a spectrophotometer at 620 nm.

For blood tests (except for the blood glucose measurements), blood was taken by cardiac puncture from mice being anesthetized with ketamine/xylazine prior to the procedure. 

For pyruvate and glucose tolerance tests, mice were fasted for 16 h for pyruvate tolerance test (PTT) and for 6 h for glucose tolerance test (GTT). Pyruvate (2 mg/g) or glucose (2 mg/g) were diluted in PBS and injected intraperitoneally. Blood samples were taken from the tail vein, and glucose was measured using a blood glucose monitor. 

Animals underwent a whole body MRI (7 tesla MRI small animal, quadrature volume coil 40 mm inner diameter, Bruker PharmaScan^®^; Bruker, Billerica, MA, USA), under isoflurane anesthesia and the monitoring of the respiratory frequency. The following axial high-resolution Turbo-RARE T2-weighted sequence was performed for imaging the whole body of the animals: TR: 4150 ms; TE: 36 ms; Fip angle: 180°; Matrix: 256 × 256; FOV 40 mm; slice thickness: 2 mm; Nex: 5.

### 4.2. Quantitative PCR

Total RNA was purified from liver tissue using NucleoSpin RNA isolation kit (Macherey-Nagel, Düren, Germany). Samples were quality-controlled using a Nanodrop 2000 (Thermo Fisher, Waltham, MA, USA). Then, 1 µg of total RNA was reverse transcribed into cDNA using a High Capacity RNA-to-cDNA Kit (Applied Biosystems, Waltham, MA, USA). Quantitative PCR (QPCR) was performed using Fast SYBR Green PCR Master Mix (Applied Biosystems). QPCR was carried out using a 7500 fast Real-Time PCR system (Applied Biosystems) and the specific primers listed in the Table 1. 

Fold-changes were calculated using the comparative −ΔΔCT method using 18S rRNA as a reference for normalization. Results are from 7 control and 9 dnPKA^iEC^ mice (three separate experiments/litters).

### 4.3. Immunostaining

*For liver sections:* Livers were fixed with 4% PFA overnight at 4 °C and cut into 200 μm sections by vibratome. Slices were blocked and permeabilized in TBST (TBS with 0.1% Triton X-100) and 5% BSA for 2 h at room temperature. The following primary antibodies were used for staining: anti-Endomucin (clone: V.7C7; Cat. #: 14-5881-85; Fisher Scientific, Waltham, MA, USA), anti-CD31 (Abcam; Cat. #: ab28364), anti-LYVE1 (AngioBio; #11-034), anti-Glutamine Synthetase (ab49873; Abcam, Cambridge, UK), FITC-coupled anti-SMA (Sigma; #: F3777), Alexa Fluor-488-coupled anti-GFP (Thermo Fisher; #A-21311), Alexa Fluor-568-coupled Isolectin B4 (Thermo Fisher; #: I21412). For visualization, Alexa Fluor 488, 555 or 647-coupled secondary antibodies (Thermo Fisher; 1/500) were used. Tissues were washed and mounted on slides in fluorescent mounting medium (Dako GmbH, Jena, Germany). Images were acquired using a Leica TCS SP8 confocal microscope.

*For skin and diaphragm whole mount staining:* Dissected skin or diaphragms were fixed with 4% PFA overnight at 4 °C. After permeabilization and blocking with PBST (PBS with 0.1% Triton X-100) and 5% BSA for 2 h at room temperature, primary antibodies diluted in PBST/BSA were added and incubated overnight at 4 °C. After washing with PBST, secondary antibodies (Thermo Fisher; 1/500) were added and incubated for 2 h at room temperature. Skin or diaphragms were mounted between a slide and coverslip using fluorescent mounting medium (Dako). 

*For fat pads staining:* Fat was removed by dehydration (serial washes in 25%, 50%, 75% and 100% ethanol; each for 20 min; diluted with PBST), which was followed by a treatment of dehydrated tissue with Histo-Clear II (National Diagnostics, Atlanta, GA, USA) for 30 min. Tissue was rehydrated by a reverse series of decreasing concentrations of ethanol. The staining of blood vessels with isolectin B4 was performed as described previously [42].

### 4.4. Western Blotting

For Western blotting, frozen tissue was pulverized in liquid nitrogen and homogenized in 50 mM Tris (pH 7.4), 150 mM NaCl, 1 mM EDTA, 1% Triton X-100, 0.1% SDS containing Complete EDTA-free Protease Inhibitor Cocktail (Sigma) and PhosSTOP phosphatase inhibitor cocktail (Sigma). Tissue homogenates were spun down @ 4 °C (16,000 g for 10 min) to remove debris. Protein concentration was determined using a Pierce BCA Protein Assay Kit (Thermo Fisher). Homogenates were diluted with SDS sample buffer and boiled for 5 min. Equal amounts of protein were loaded and separated by SDS-PAGE using a Criterion electrophoresis system (Bio-Rad Laboratories, Hercules, CA, USA), transferred onto the polyvinylidene fluoride membrane, and immunoblotted with specific antibodies: rabbit anti-albumin (Bio-Rad; #AHP1478); rabbit anti-Pck1 (Abcam; ab28455); or rabbit anti-GAPDH (#2118; Cell Signaling Technologies, Danvers, MA, USA).

### 4.5. Scanning Electron Microscopy

For scanning electron microscopy (SEM), mice were perfused with PBS followed with 2.5% glutaraldehyde in 0.1 M cacodylate buffer pH 7.2. After perfusion, the liver was removed and cut in 1–2 mm thick slices and further fixed overnight at 4 °C in fresh fixative of the same composition. Subsequently, samples were washed with cacodylate buffer, osmicated with 1% osmiumtetroxide for 2 h, washed in H_2_O and dehydrated in a graded ethanol series. Finally, the samples were critical point dried in a Leica CPD300 apparatus, mounted with silver paint on support stubs and coated with 4 nm chromium in a Leica ACE600 coating machine. SEM images were taken with a Zeiss Sigma FESEM at 5 kV acceleration voltage at variable pressure of 20 Pa and a nominal magnification of 20,000×. Quantification of the liver sinusoids porosity was performed as described in [43]. 

### 4.6. Statistical Analysis

Statistical analyses were performed using Prism 7.0 (GraphPad Software, San Diego, CA, USA). The two-tailed unpaired Student’s t test was used to compare two different groups (Figure 1B, Figure 3C–E, Figure 5A–H, Figure 6B and Figure 8B,C). In cases when more than two groups were compared, one-way ANOVA test was performed, which was followed by Tukey’s multiple comparison test (Figure 8A, Figure 11 and Figure 13). In cases when control and the dnPKA^iEC^ mice were compared over a time-period, two-way ANOVA followed by Sidak’s multiple comparison test was used (Figure 1E, Figure 6C,D and Figure 7A,B). Statistically significant changes are indicated in the figure legends.

## Figures and Tables

**Figure 1 ijms-23-11419-f001:**
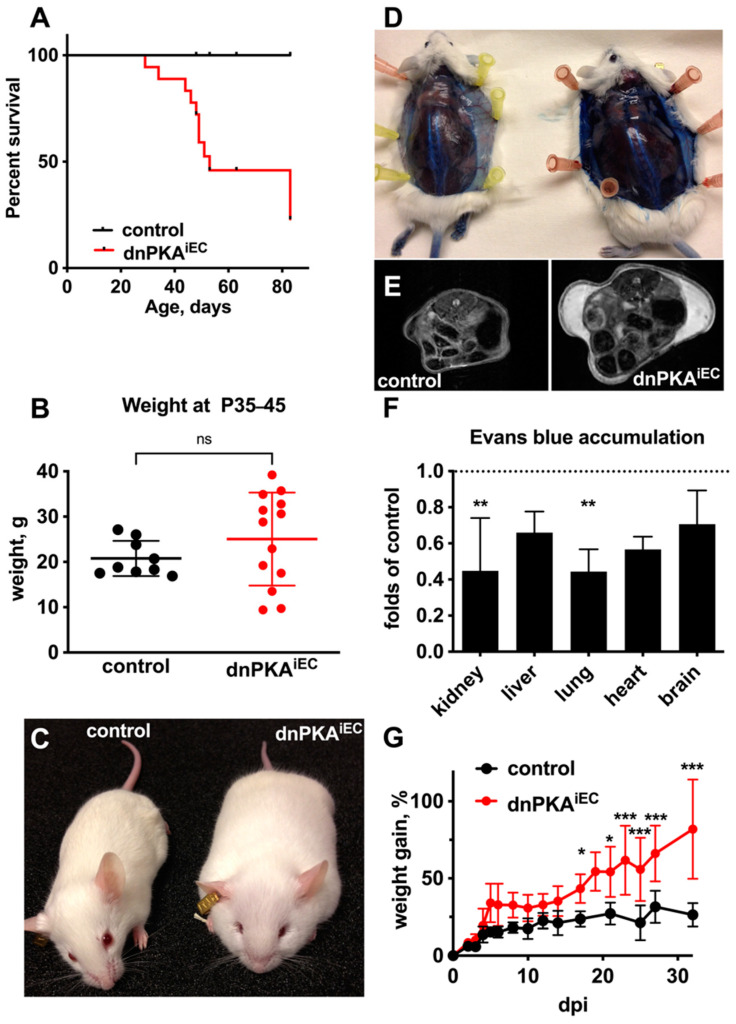
Inhibition of endothelial PKA results in subcutaneous edema and premature death. (**A**–**E**) dnPKA^iEC^ mice and their control (Tg(*Cdh5-cre/ERT2*)^1Rha^ without floxed transgenes) littermates were injected with 50 mg tamoxifen at postnatal days 1, 2 and 3. (**A**), Inhibition of endothelial PKA results in premature death. In total, 15 control and 18 dnPKA^iEC^ mice from 4 different litters were analyzed. (**B**), dnPKA^iEC^ mice have broader weight distribution than their control littermates, with some gaining more weight and some failing to gain weight normally. Here, 9 control and 13 dnPKA^iEC^ mice from 4 different litters were analyzed. Data are presented as scatter blot with individual values, means and SD. Data were analyzed using two-tailed unpaired Student’s *t*-test. ns, not significant. (**C**), dnPKA^iEC^ mice (**right**) develop a clearly visible subcutaneous edema. Presented mice are female littermates, 64 days old. (**D**), Control (**left**) and dnPKA^iEC^ (**right**) mice were injected with Evans blue and dissected after an hour. While there was no subcutaneous leakage of Evans blue in control mice, dnPKA^iEC^ mice accumulated tracers under the skin. (**E**), MRI images of control and dnPKA^iEC^ mice demonstrating subcutaneous edema. (**F**), Quantification of Evans blue in inner organs demonstrates that there was no increased accumulation of the tracer in any organ in dnPKA^iEC^ mice compared to their control littermates. Here, 3 control and 3 dnPKA^iEC^ male mice at the age of 49 days were analyzed. Shown are means ± SD. Data were analyzed using one-way ANOVA followed by Tukey’s multiple comparison test. **, *p* < 0.01. (**G**), dnPKA^iEC^ mice quickly developed edema even if injected with tamoxifen at the age of 28–32 days. Here, 20 control and 23 dnPKA^iEC^ mice were analyzed. dpi, days post injection. Please note that since mice in the panel (**F**) are injected at the age of 28–32 days, dpi values in the panel (**F**) do not correspond to age in the panels (**A**–**E**) (mice were injected at the age of 1–3 days). Shown are means ± SD. dpi, days post injection. Data were analyzed using two-way ANOVA followed by Sidak’s multiple comparison test. *, *p* < 0.05; ***, *p* < 0.001.

**Figure 2 ijms-23-11419-f002:**
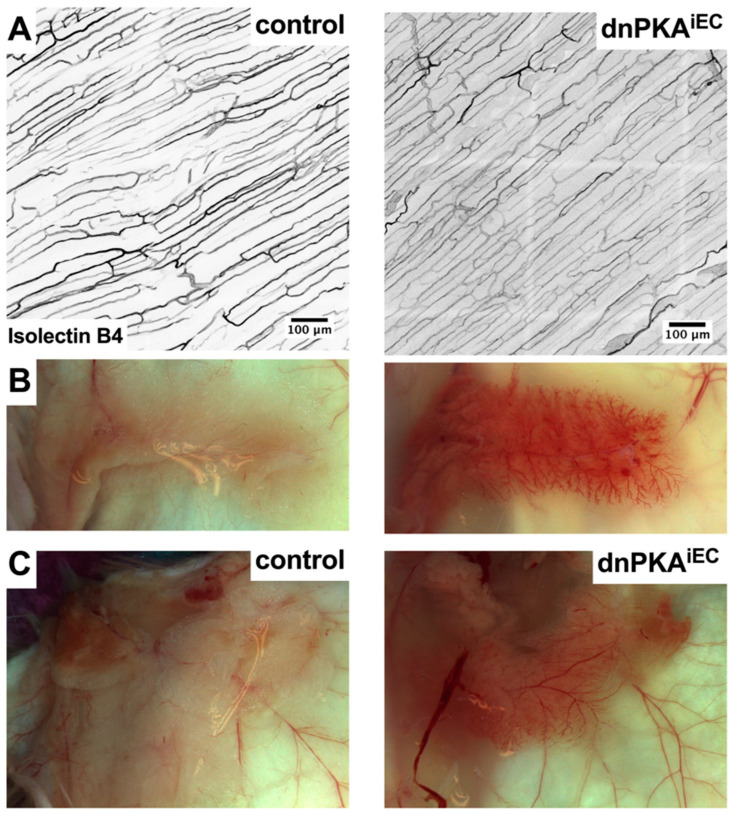
Inhibition of endothelial PKA leads to tissue-specific hypersprouting. (**A**), Isolectin B4 staining of blood vessels in dorsal skin of control (Tg(*Cdh5-cre/ERT2*)^1Rha^ without floxed transgenes) and dnPKA^iEC^ mice demonstrates no differences in morphology or density of cutaneous vessels. (**B**,**C**), Inguinal (**B**) and thoracic (**C**) fat pads of dnPKA^iEC^ mice (both in males and females) had much denser vasculature with excessive sprouting (**B**) and dilated vessels (**C**). Mice were injected with tamoxifen at the age of 28–32 days; shown are tissues and staining of 70-day-old male mice.

**Figure 3 ijms-23-11419-f003:**
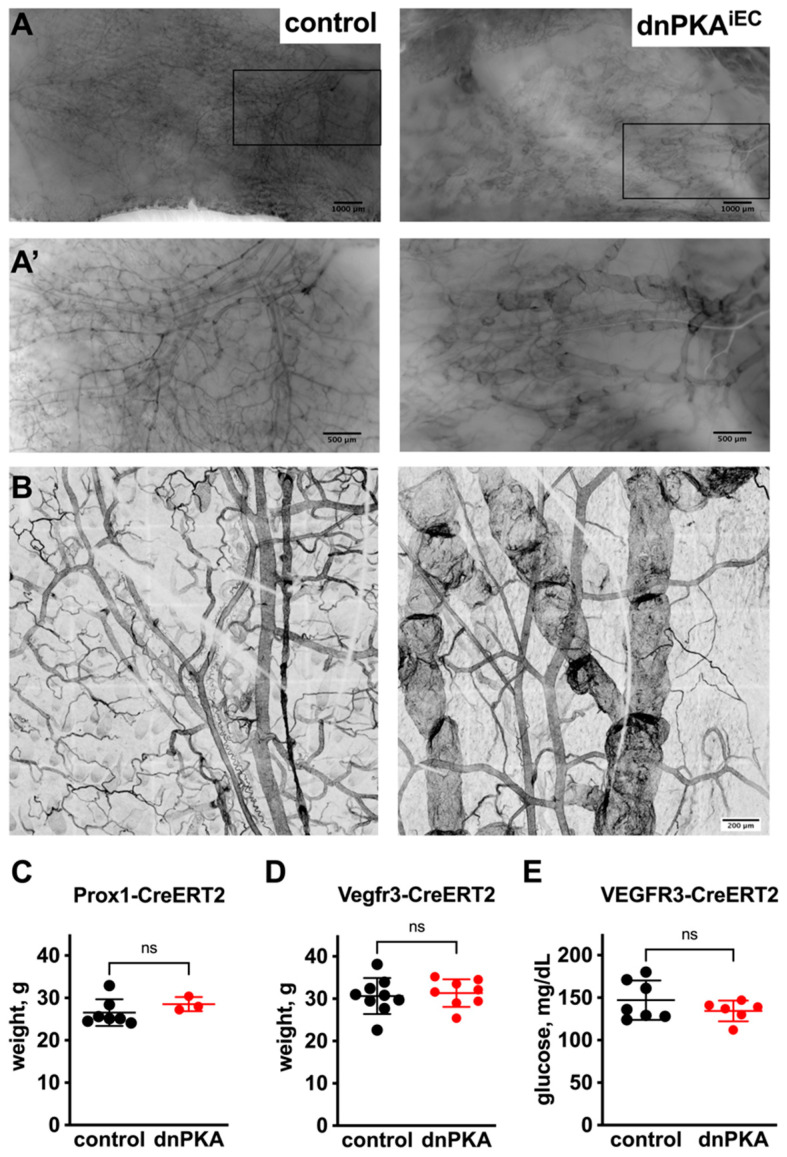
Inhibition of endothelial PKA leads to enlargement of cutaneous lymphatic vessels. (**A**), ventral skin underneath inguinal fat pad was stained with isolectin B4 to visualize blood and lymphatic (latter were identified by the presence of valves) vessels. Scale bar, 1000 µm. (**A’**), magnified image of the area marked in A shown. Strong enlargement of lymphatic vessels and reduced density of blood vessels in dnPKA^iEC^ mice is clearly visible. Scale bar, 500 µm. (**B**), projections of confocal images of cutaneous vessels in control (Tg(*Cdh5-cre/ERT2*)^1Rha^ without floxed transgenes) and dnPKA^iEC^ littermates. Scale bar, 200 µm. (**C**,**D**), inhibition of PKA in lymphatic endothelial cells induced by either Prox1-CreERT2 (**C**) or Vegfr3-CreERT2 (**D**) was not sufficient to provoke edema. In addition, inhibition of PKA induced by Vegfr3-CreERT2 was not sufficient to provoke hypoglycemia (**E**). For (**A**,**B**), mice were injected with tamoxifen at the age of 28–32 days and dissected at the age of 72 days. For (**C**–**E**), mice were injected with tamoxifen at postnatal days 1–3; shown are weights at the age of 88 days (for Prox1-CreERT2; panel (**C**)), 127 days (for Vegfr3-CreERT2; panel (**D**)) and blood glucose levels between 138 and 167 days (panel (**E**)). For (**C**), 7 control (Prox1-CreER^T2^ without floxed transgenes) and 3 mutant (carrying both Prox1-CreER^T2^ and *Prkar1a^tm2Gsm^* alleles) mice were analyzed; for (**D**), 9 control (Vegfr3-CreER^T2^ without floxed transgenes) and 8 mutant (carrying both Vegfr3-CreER^T2^ and *Prkar1a^tm2Gsm^* alleles) mice were analyzed; for (**E**), 7 control (Vegfr3-CreER^T2^ without floxed transgenes) and 6 mutant (carrying both Vegfr3-CreER^T2^ and *Prkar1a^tm2Gsm^* alleles) mice were analyzed. Data are presented as scatter plot with individual values, means ± SD. Data were analyzed using two-tailed unpaired Student’s *t*-test. ns, not significant.

**Figure 4 ijms-23-11419-f004:**
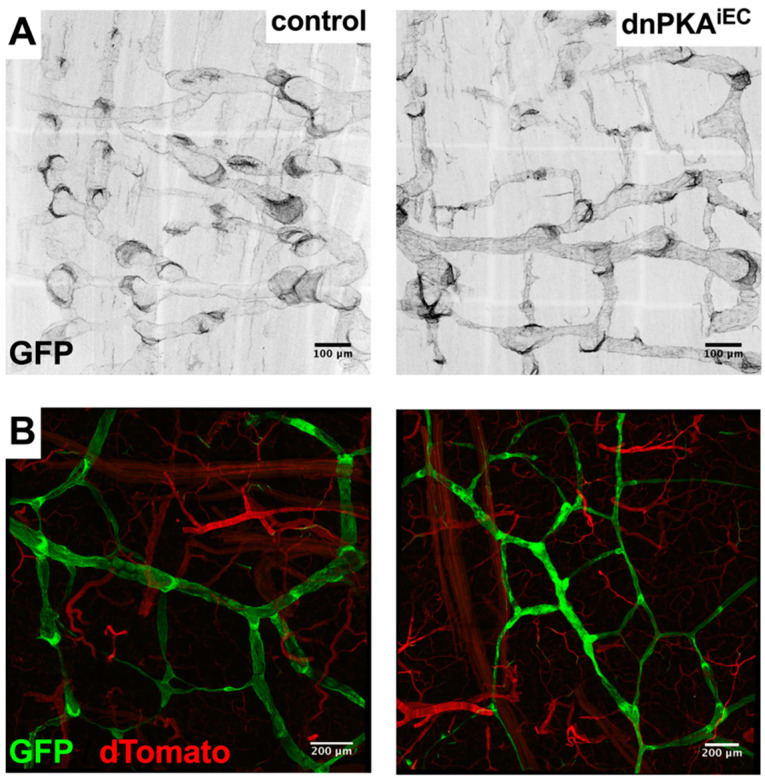
Expression of dnPKA in lymphatic endothelial cells does not induce changes in lymphatic morphology. dnPKA expression in lymphatic endothelial cells was induced by Vegfr3-CreERT2 (tamoxifen injections at postnatal days 1 to 3). Successful recombination in the lymphatics of diaphragm (**A**) and ventral skin (**B**) were monitored by GFP expression from the mT/mG reporter. Expression of dnPKA in lymphatic endothelial cells only did not induce any changes in the density or morphology of lymphatic vessels. Scale bar, 100 µm in (**A**) and 200 µm in (**B**). Control (Vegfr3-CreER^T2^ carrying mT/mG transgene) and mutant (carrying Vegfr3-CreER^T2^, *Prkar1a^tm2Gsm^*, and mT/mG alleles) mice were injected with tamoxifen at the age of 1–3 days; shown are tissues and staining of 74-day-old female mice.

**Figure 5 ijms-23-11419-f005:**
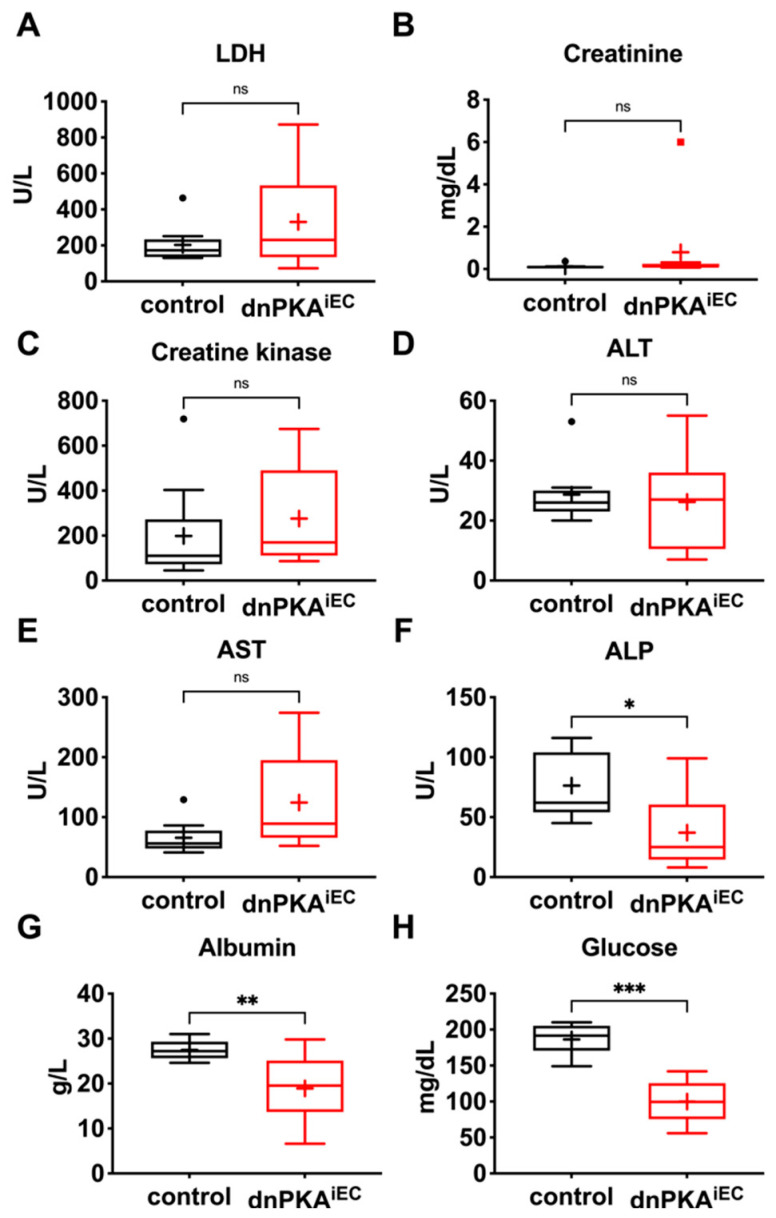
Inhibition of endothelial PKA results in hypoalbuminemia and hypoglycemia. Blood samples from dnPKA^iEC^ mice and control (Tg(Cdh5-cre/ERT2)1Rha without floxed transgenes) littermates were analyzed for markers of tissue damage: (**A**), lactate dehydrogenase; (**B**), creatinine; (**C**), creatine kinase; (**D**), alanine aminotransferase; (**E**), aspartate aminotransferase; (**F**), alkaline phosphatase; (**G**), albumin; and (**H**), glucose levels. In total, 9 control and 10 dnPKA^iEC^ mice from 4 different litters were analyzed. Mice (males and females) were injected with tamoxifen at the age of 30–34 days, and blood was taken by cardiac puncture at the age of 68–93 days. Data are presented as box and whiskers plot. Means are indicated with a plus. Data were analyzed using two-tailed unpaired Student’s *t*-test. ns, not significant; *, *p* < 0.05; **, *p* < 0.01; ***, *p* < 0.001.

**Figure 6 ijms-23-11419-f006:**
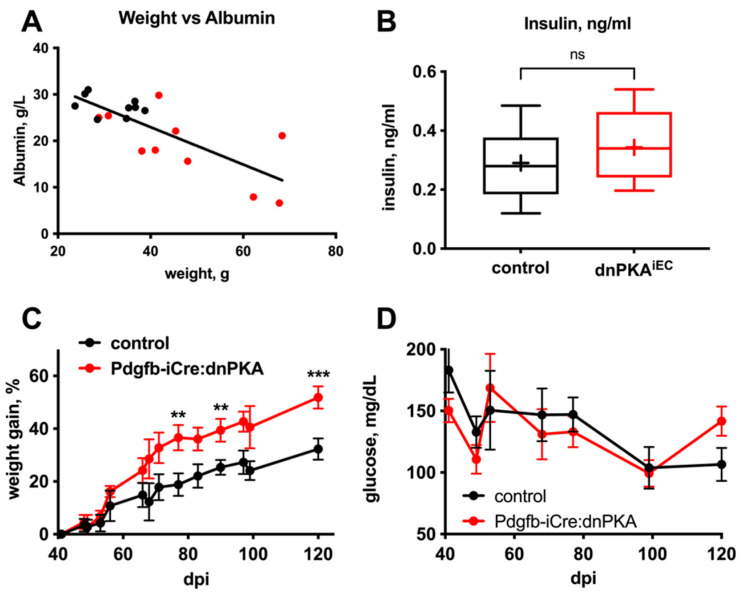
(**A**) Plasma albumin levels showed a negative correlation with the degree of edema (manifested in a weight gain) in dnPKA^iEC^ mice was observed. (**B**) No differences in blood insulin levels were detected between ad libitum fed dnPKA^iEC^ mice and their control (Tg(*Cdh5-cre/ERT2*)^1Rha^ without floxed transgenes) littermates. Here, 9 control (black) and 10 dnPKA^iEC^ (red) mice from 4 different litters were analyzed. Mice (males and females) were injected with tamoxifen at the age of 30–34 days, and blood was taken by cardiac puncture at the age of 68–93 days. Data presented as box and whiskers plot. Means are indicated with a plus. Data were analyzed using two-tailed unpaired Student’s *t*-test. (**C**,**D**) Induction of dnPKA specifically in blood vessel endothelial cells using Pdgfb-iCre led to a much slower and modest development of edema than in dnPKA^iEC^ (**C**) and no hypoglycemia (**D**). Here, 10 control (Tg(*Pdgfb-icre/ERT2*)^1Frut^ without floxed transgenes; black) and 5 mutant (red) mice were analyzed (for (**C**)). For (**D**), 5 control (black) and 3 mutant (red) mice were analyzed. For panels (**C**,**D**), mice (males and females) were injected with tamoxifen at the age of 1–3 days weights, and blood glucose levels were measured starting with the day 40 after injection (40 dpi). dpi, days post injection. Data (**C**,**D**) were analyzed using two-way ANOVA followed by Sidak’s multiple comparison test. ns, not significant; **, *p* < 0.01; ***, *p* < 0.001.

**Figure 7 ijms-23-11419-f007:**
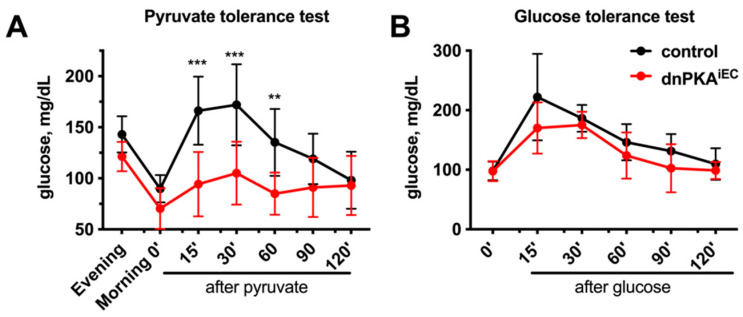
Inhibition of endothelial PKA impairs gluconeogenesis. (**A**), dnPKA^iEC^ mice were unable to respond to a bolus i.p. injection of pyruvate by increasing blood glucose levels. Here, 10 control (Tg(Cdh5-cre/ERT2)1Rha without floxed transgenes) and 15 dnPKA^iEC^ mice were analyzed. Shown are means ± SD. (**B**), dnPKA^iEC^ mice preserved the ability to regulate blood glucose levels after a bolus i.p. injection of glucose. 5 control (Tg(Cdh5-cre/ERT2)1Rha without floxed transgenes) and 6 dnPKA^iEC^ mice were analyzed. Shown are means ± SD. Mice were injected with tamoxifen at the age of 28–32 days. Pyruvate and glucose tolerance tests were performed 30–40 days after tamoxifen injection (at the age between 60 and 70 days). Data were analyzed using two-way ANOVA followed by Sidak’s multiple comparison test. **, *p* < 0.01; ***, *p* < 0.001.

**Figure 8 ijms-23-11419-f008:**
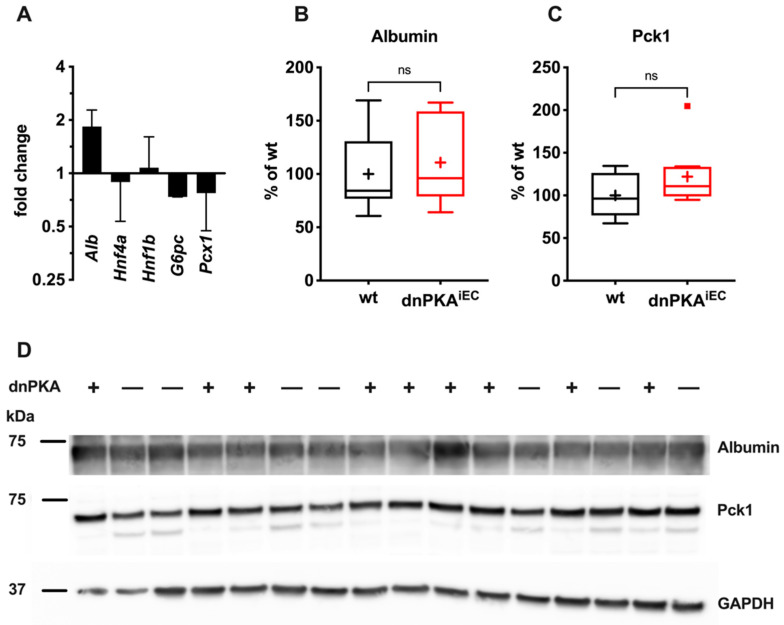
Hepatic marker expression is not decreased under inhibition of endothelial PKA. (**A**), Albumin (*Alb*), *Hnf4a* and *Hnf1b*, *G6pc* and *Pcx1* mRNA levels were analyzed in control and dnPKA^iEC^ mice. Protein levels of albumin (**B**) and phosphoenolpyruvate carboxykinase (Pck1; (**C**)) were analyzed by Western blotting. (**D**) Blots for albumin, Pck1 and GAPDH are shown. Here, 7 control (Tg(Cdh5-cre/ERT2)1Rha without floxed transgenes) and 9 dnPKA^iEC^ mice from three different litters were analyzed. Mice (males and females) were injected with tamoxifen at the age of 30–35 days and dissected at the age of 61–84 days. Shown are means ± SD. Data were analyzed using one-way ANOVA followed by Tukey’s multiple comparison test (**A**) or two-tailed unpaired Student’s *t*-test (**B**,**C**). ns, not significant.

**Figure 9 ijms-23-11419-f009:**
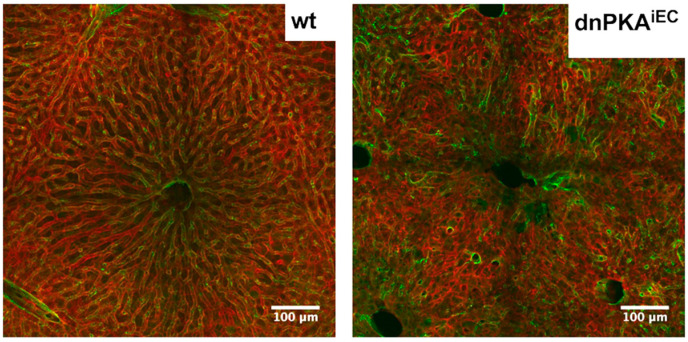
Disorganization of the hepatic vasculature in dnPKA^iEC^ mice. Representative liver slices of control (Tg(Cdh5-cre/ERT2)1Rha carrying mT/mG transgene) and dnPKA^iEC^ mice carrying the mT/mG reporter are shown. Endothelial cells are green due to expression of GFP, non-endothelial cells are red due to expression of dTomato. Scale bar, 100 µm.

**Figure 10 ijms-23-11419-f010:**
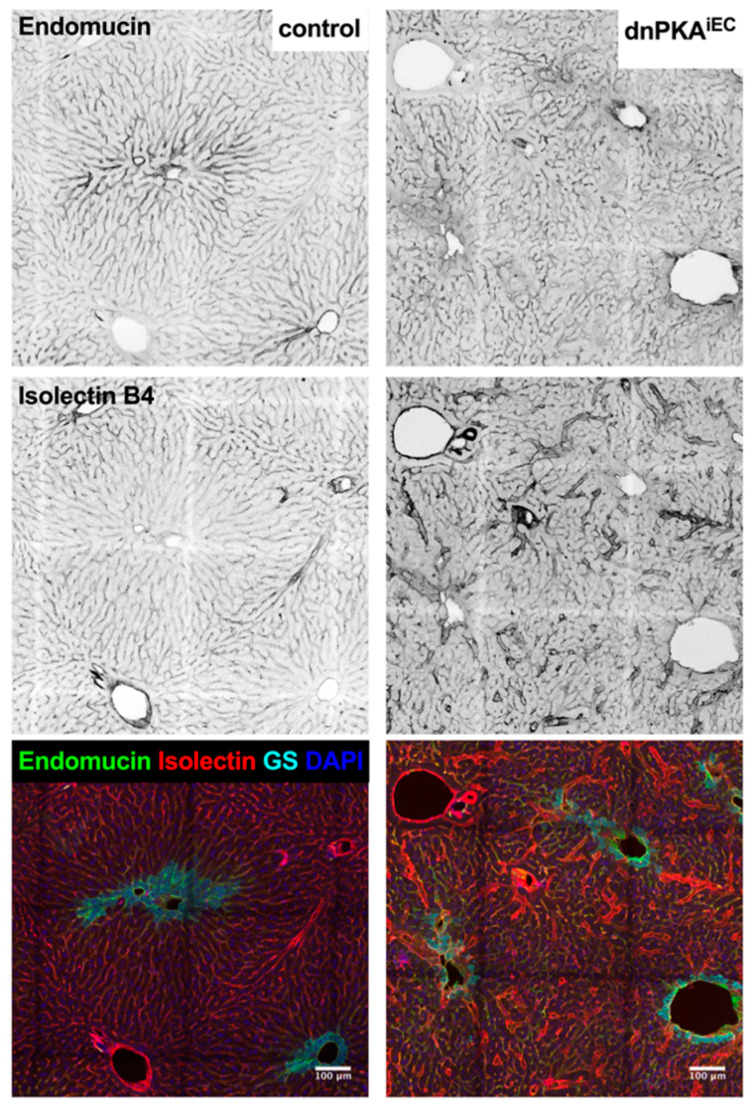
Hepatocyte zonation is preserved in dnPKA^iEC^ mice. Liver slices of control (Tg(Cdh5-cre/ERT2)1Rha without floxed transgenes) and dnPKA^iEC^ mice were stained for endomucin (**upper** panels) or isolectin B4 (**middle** panels). Expression of endomucin-positive cells and cells stained with isolectin B4 demonstrate remarkable spatial distribution in control mice (**left** panels) with high levels of endomucin and low levels of isolectin B4 staining around the central vein (identified by expression of glutamine synthetase in hepatocytes surrounding the central vein, GS; cyan in **lower** panels). Disorganization of the hepatic vasculature in dnPKA^iEC^ mice was evident throughout all liver zones (**right** panels). Scale bar, 100 µm. Mice were injected with tamoxifen at the age of 28–32 days and dissected at the age of 67–72 days.

**Figure 11 ijms-23-11419-f011:**
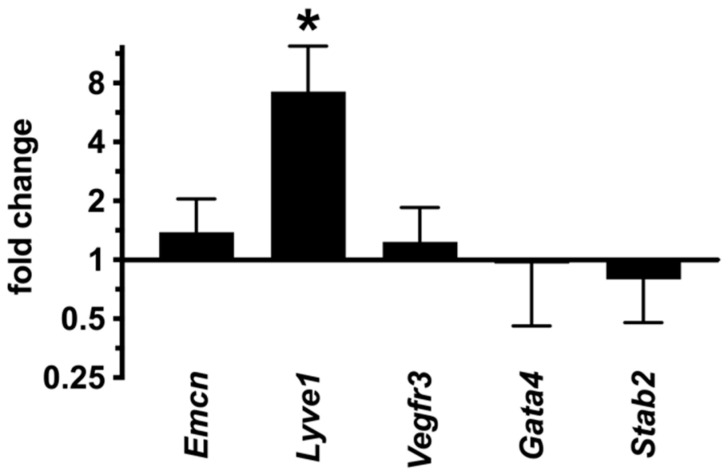
*Lyve1* expression in the liver of dnPKA^iEC^ mice was increased while other sinusoidal markers tested (*Emcn*, *Vefgr3*, *Gata4* and *Stab2*) were unchanged. Mice (males and females) were injected with tamoxifen at the age of 30–35 days and dissected at the age of 61–84 days. Here, 7 control (Tg(*Cdh5-cre/ERT2*)^1Rha^ without floxed transgenes) and 9 dnPKA^iEC^ mice from three different litters were analyzed. Shown are means ± SD. Data were analyzed using one-way ANOVA followed by Tukey’s multiple comparison test. *, *p* < 0.05.

**Figure 12 ijms-23-11419-f012:**
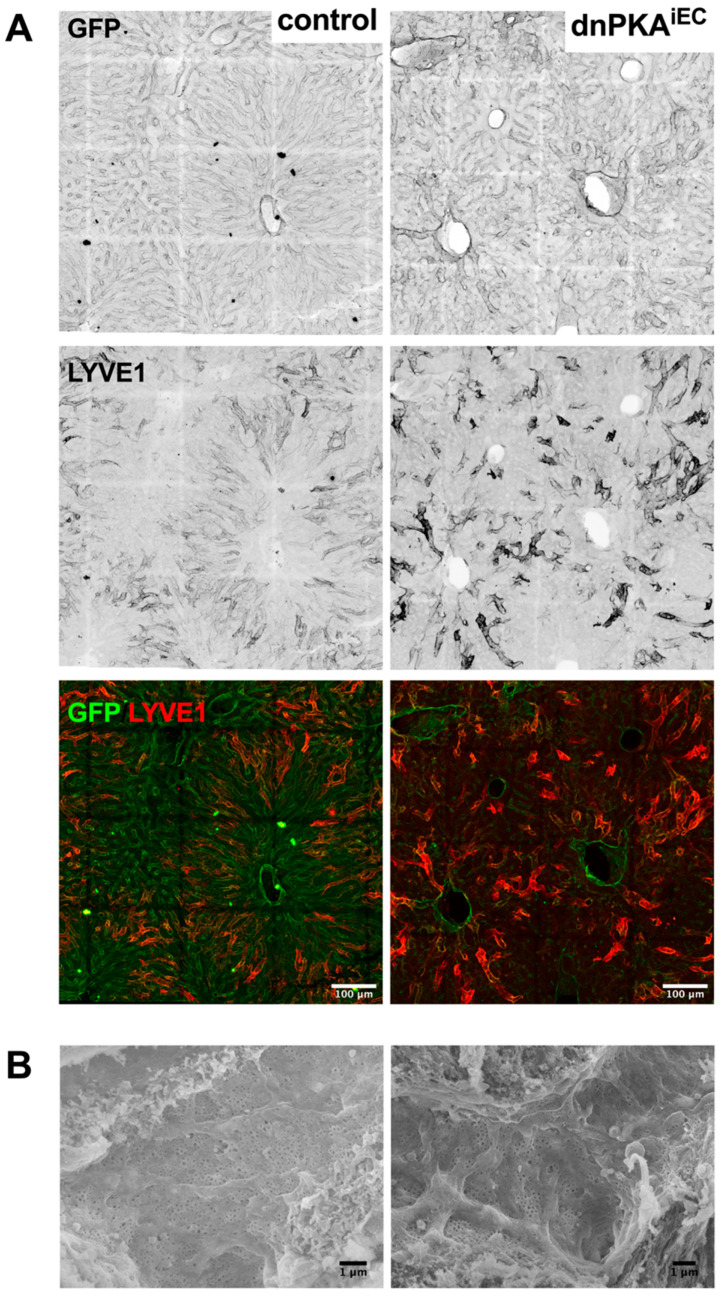
Liver sinusoidal cells in dnPKA^iEC^ mice preserve their biochemical and morphological features (**A**), despite disorganization of the hepatic vasculature, LYVE1, a marker of liver sinusoidal cells (LSECs) was expressed and even increased in dnPKA^iEC^ mice. Scale bar, 100 µm. (**B**), LSECs in dnPKA^iEC^ mice also preserved fenestration, which is a feature that is lost during the pathological dedifferentiation of hepatic vasculature. Porosity of liver sinusoids (see Materials and Methods section) was 3.8 ± 0.3% (n = 5) for control mice and 4.1 ± 1.7% (n = 4) for dnPKA^iEC^ mice. Scale bar, 1 µm. Control (Tg(Cdh5-cre/ERT2)1Rha carrying mT/mG transgene) mice and dnPKA^iEC^ mice carrying the mT/mG reporter were injected with tamoxifen at the age of 28–32 days and dissected at the age of 65–74 days.

**Figure 13 ijms-23-11419-f013:**
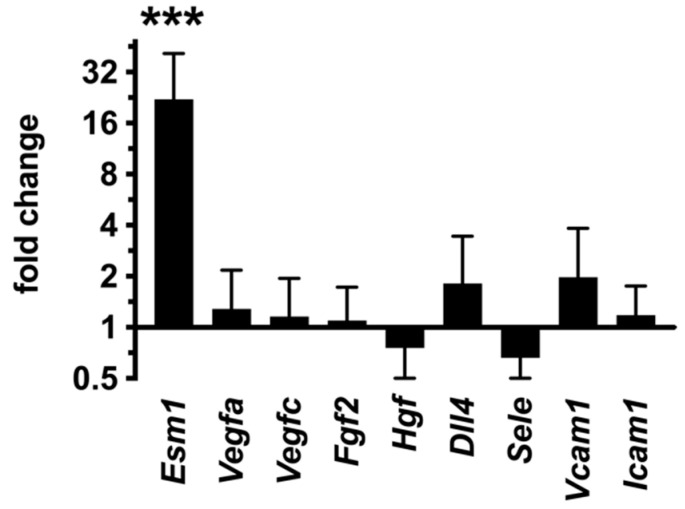
Expression of the tip cell marker *Esm1* is increased in dnPKA^iEC^ mice. While the expression of *Esm1* was strongly increased in the liver of dnPKA^iEC^ mice, none of the other tested angiogenic factors (*Vegfa*, *Vegfc*, *Fgf2*, *Hgf* and *Dll4*) was affected. In addition, markers of inflammatory-activated endothelium (*Sele*, *Vcam1* and *Icam1*) were not increased. Mice (males and females) were injected with tamoxifen at the age of 30–35 days and dissected at the age of 61–84 days. Here, 7 control (Tg(*Cdh5-cre/ERT2*)^1Rha^ without floxed transgenes) and 9 dnPKA^iEC^ mice from three different litters were analyzed. Shown are means ± SD. Data were analyzed using one-way ANOVA followed by Tukey’s multiple comparison test. ***, *p* < 0.001.

**Table 1 ijms-23-11419-t001:** QPCR primers.

Target	Forward Primer (5′-3′)	Reverse Primer (5′-3′)
*Alb*	GCAGATGACAGGGCGGAACTTG	CAGCAGCAATGGCAGGCAGAT
*Hnf4a*	GCTGTCCTCGTAGCTTGACC	TTAAGAAGTGCTTCCGGGCT
*Hnf1b*	AGGGAGGTGGTCGATGTCA	TCTGGACTGTCTGGTTGAACT
*G6pc*	GTCTTGTCAGGCATTGCTGTG	CAGGTAGAATCCAAGCGCGA
*Pcx1*	GGGCGGAGCTAACATCTACC	TATACTCCAGACGCCGGACA
*Emcn*	GGTTCCCAGAACAAGACTGAGA	TGGAATAGGAGGGGGTGGTT
*Lyve1*	CACTAGGCACCCAGTCCAAG	TCCAACCCATCCATAGCTGC
*Vegfr3*	CACAATCCCCATGCTCTGGT	TGTGAGGAGGCACATTCACC
*Gata4*	CCATCTCGCCTCCAGAGT	CTGGAAGACACCCCAATCTC
*Stab2*	CCAGGGATATCCAGGACGTA	TGTCCAGACGGCTACATCAA
*Esm1*	GCAAGAGGACAGTGCTGGAT	GGTGCCATAGGGACAGTCTTT
*Vegfa*	AGGAGCCGAGCTCATGGA	GGGACCACTTGGCATGGTG
*Vegfc*	TGTGCTTCTTGTCTCTGGCG	CCTTCAAAAGCCTTGACCTCG
*Fgf2*	TTCATAGCAAGGTACCGGTTG	AGAAGAGCGACCCACACG
*Hgf*	GCGAATTGGTGTTCTGCCTG	GAGCAGACTGATCCCTAAAGCA
*Dll4*	GGCAAACTGCAGAACCACAC	TGGCTTCTCACTGTGTAACCG
*Sele*	TCTATTTCCCACGATGCATTT	CTGCCAAAGCCTTCAATCAT
*Vcam1*	CTGGGAAGCTGGAACGAAGT	GCCAAACACTTGACCGTGAC
*Icam1*	TTTGAGCTGAGCGAGATCGG	AGAGGTCTCAGCTCCACACT

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
