# Peer review of "Vascular and Liver Homeostasis in Juvenile Mice Require Endothelial Cyclic AMP-Dependent Protein Kinase A"

_ijms, 2022, doi:10.3390/ijms231911419_

Round 1

Reviewer 1 Report

The manuscript by Nedvetsky et al. aimed to investigate the effect of dominant-negative PKA on the vascular homeostasis. This study was of interest. However, there are several deficits needing to be addressed.

1.      The scientific writing could be improved. There are many errors in grammar in the current form of manuscript. An English editor is highly recommended.

2.      The evidence for supporting the subcutaneous edema was not convincing; please provide histological examination with H&E staining.

3.      Can the disorganization of vascular structure be found in other organs? Particularly, the results of vascular structure in the heart and kidney, two important organs for the body fluid homeostasis should be provided.

4.      Can such pathological changes in liver vasculature be found in human diseases?

5.      The results of statistical analysis were not consistent, please reorganize these data. The label for the significance of statistical analysis needed to be defined in figure legend.

6.      Was there significance in body weight or weight gain in two groups of figure 1 and 6?

7.      The label of significance of statistical analysis was not clear in figure 5, 7 and 11. What’s the definition of stars?

8.      The mechanism underlying the pathological changes in liver vasculature is still unclear, the in-depth discussion is required for this manuscript. 

9.      In figure 8, what’s the significance of statistical analysis in mRNA expression? In addition, the protein expression for albumin in the liver should be examined.

Reviewer 2 Report

Nedvetsky and colleague investigated the involvement of  endothelial PKA on postnatal vascular homeostasis using CreERT2-mediated dominant negative PKA expression. They demonstrated that inhibition of PKA  induces excess blood vessels, resulting in subcutaneous edema, liver abnormalities and premature death. Therefore, they concluded that endothelial PKA is necessary for maintaining the  postnatal angiogenic homeostasis by preventing excess blood vessel formation. This study contains various interesting  insights for the functions of PKA in adult endothelial cells. Only some miner correction is necessary.

1. Are there any data that show the extent of PKA inhibition though the expression of dnPKA in endothelial tissues?

2. In section 2.5, the authors should show the data on the expression of gluconeogenesis-related genes, if they investigated.

3. It is better section 2.7 is moved to Supplemental materials, since significant results were hardly gained.

4. Line546- It is necessary to show the approval no. for the animal experiments.

Round 2

Reviewer 1 Report

Authors have addressed most of my comments. I have no further comment.